# Peer review of "Chemometric Classification of Colombian Cacao Crops: Effects of Different Genotypes and Origins in Different Years of Harvest on Levels of Flavonoid and Methylxanthine Metabolites in Raw Cacao Beans"

_molecules, 2022, doi:10.3390/molecules27072068_

Round 1

Reviewer 1 Report

Submitted manuscript presents interesting study focused on differentiation of various Colombia cocoa beans (types of clones) based on the levels of selected chemical markers.

English: In spite of the fact I do not feel qualified to evaluate language I have noticed several mistakes in grammar / spelling

References: In several parts on the manuscript, references are missing, even if it is mentioned in the text

  • Rows 87, 88 – „few studies“…
  • Row 117 - …various authors,…

Fig 2. – It is not clear to me what exactly letters a/b/c/d mean. It is only stated there that “the same letters indicate that no significant difference”… ; the same comment applies to the Fig. 4

Row 159 – This statement does clearly not correspond with the numbers in the pictures and cannot be used as general statement for all cocoa clones. In addition, instead of “catechins”, probably “catechin monomers” should be used

Row 166 – 1.70 mg/g – is this number correct? Is not it e.g. average instead of the highest concentration?

Row 186 – harvest conditions – could it be a bit more described there or in the paragraph 3.2?

Row 192 – “products” – which types of products authors have in mind if paper is focused on raw beans? Raw beans itself can be considered as “products” (paper deals with raw cocoa beans), but these may be also processed products such as cacao, chocolate etc. In following sentence, effectiveness and safety are mentioned – it is a question whether in case of markers monitored in submitted manuscript it makes sense to discuss these characteristics

Row 251 - 3.2 Samples – sampling scheme from the whole plantation / sampling site should be better described; it can be considered to add list of samples and concentrations of individual markers as supplementary material for readers

Row 275 – 3.4 Determination of … - chromatograms should be also added to document separation and identification of individual markers

Row 316-320 – I have only comment on follow-up processing of cocoa beans for further production of chocolates and other types of cocoa-based products. This processing can influence content of monitored markers. It there any follow-up study planned to elucidate it?

Author Response

Response to Reviewer 1 Comments

Submitted manuscript presents interesting study focused on differentiation of various Colombia cocoa beans (types of clones) based on the levels of selected chemical markers.

Response 1: The authors appreciate the reviewer's comments, and then we proceed to explain, point by point, the changes made.

English: In spite of the fact I do not feel qualified to evaluate language I have noticed several mistakes in grammar / spelling

Response 2: The english of this version of the manuscript has been technically reviewed by American Journal Expert. The editorial certificate for the manuscript is attached.

References: In several parts on the manuscript, references are missing, even if it is mentioned in the text

  • Rows 87, 88 – „few studies“…
  • Row 117 - …various authors,…

Response 3: This version of the manuscript has been modified in these sections. In some cases (for example, Rows 87, 88 – "few studies“) the non-existence of studies is reported. In other cases (for example, Row 117 -"various authors"), pertinent references are included.

Fig 2. – It is not clear to me what exactly letters a/b/c/d mean. It is only stated there that “the same letters indicate that no significant difference”… ; the same comment applies to the Fig. 4.

Response 4: In this new version, the text was replaced.

In figure 2 by: “Analysis was performed using one-way ANOVA, same letters indicate non-significant difference between clones (Bonferroni’s multiple comparison test, p>0.05). The analysis includes the two harvest times.”

In figure 4 by: “Chemical composition of representative cocoa clones in different years of harvest (harvest one: 2019, harvest two: 2020). Analysis was performed using one-way ANOVA, same letters indicate non-significant difference between harvest periods for each clone (Bonferroni’s multiple comparison test, p>0.05)” in figure 4.

Row 159 – This statement does clearly not correspond with the numbers in the pictures and cannot be used as general statement for all cocoa clones. In addition, instead of “catechins”, probably “catechin monomers” should be used

Response 5: This version of the manuscript has been modified in this section, according to the instructions of the evaluator.

Row 166 – 1.70 mg/g – is this number correct? Is not it e.g. average instead of the highest concentration?

Response 6: 1.70 mg/g is the average. It was replaced by “with a maximum average content of 1.70 mg/g”.

Row 186 – harvest conditions – could it be a bit more described there or in the paragraph 3.2?

Response 7: In this version of the manuscript particular conditions for the two regions involved in the study were included.

Row 192 – “products” – which types of products authors have in mind if paper is focused on raw beans? Raw beans itself can be considered as “products” (paper deals with raw cocoa beans), but these may be also processed products such as cacao, chocolate etc. In following sentence, effectiveness and safety are mentioned – it is a question whether in case of markers monitored in submitted manuscript it makes sense to discuss these characteristics

Response 7: Indeed, as the evaluator comments, we are referring to products processed from cocoa beans. These products are mostly food products, but they could be functional products to the extent that a certain concentration of bioactive molecules is guaranteed, and the verification of their functional benefits. In this new version, the sentence is slightly modified for a better understanding of the readers

Row 251 - 3.2 Samples – sampling scheme from the whole plantation / sampling site should be better described; it can be considered to add list of samples and concentrations of individual markers as supplementary material for readers.

Response 8: The sampling scheme was completed, see item 3.2 Samples. A list of samples and concentrations of individual markers was added as supplementary material (Table 1).

Row 275 – 3.4 Determination of … - chromatograms should be also added to document separation and identification of individual markers.

Response 9: A representative chromatogram was added in Supplementary Material (Figure 6).

Row 316-320 – I have only comment on follow-up processing of cocoa beans for further production of chocolates and other types of cocoa-based products. This processing can influence content of monitored markers. It there any follow-up study planned to elucidate it?

Response 10: The answer is yes. Processing may affect the content of metabolites of interest. Our idea is to work with the cocoa clone with the highest content of bioactive compounds, and to develop a natural ingredient to enrich processed chocolate products. This will ensure the minimum content of the molecules, specifically catechin monomers.

Reviewer 2 Report

line 50-62 – this paragraph needs expansion in citations. Here I provide recommended articles: https://doi.org/10.1016/j.lwt.2018.08.030 , https://doi.org/10.1016/j.foodres.2016.03.026, https://dx.doi.org/10.1021/acs.jafc.0c00412  ,  https://doi.org/10.1080/07373937.2016.1175470

Results and Discussion -  authors need to discuss the findings with literature (chemical composition, influence of different harvest years, influence of different cultivar areas, influence of different clones). The current discussion of the results with literature is scarce.

line 119-121 - please describe better with more details the principal component analysis (PCA). Are all analyzed parameters qualified for PCA analysis? What was the lowest correlation value of parameters with the generated first and second principal components?

Figure 1 A and B – authors must enlarge numbers and codes of substances especially on figure 1B for better readability / or provide better resolution of figure like Figure 3

Figure 2 – from which year of cocoa harvest are these values? 2019 or 2020 or both?, please give information

Figure 3 – give information which harvest is from 2019 and which from 2020

For PCA and PLS-DA provide as supplementary materials figures with component 3 and 4 with commentary, because the Figure 1 explain only 55,3% of total variance and Figure 3 explain only 50,5% of total variance.

line 252-261 – authors must provide information how long gathered pods were stored and in what conditions before cocoa beans were separated from them and freeze-dried. Please provide also name of model and manufacturer of device used for grinding the beans. Whether the powdered samples were vacuum packed, modified atmosphere packed or otherwise. What packaging were these samples packaged in for storage? Did the authors mean -25C, not +25C?

line 273 - Did the authors mean -20C, not +20C?

line 276-294 – Please provide information how you obtained quantitative data about procyanidin B2 (PB2), procyanidins (D1 to D11), total monomers (TM), total procyanidins (TP), monomer/procyanidin ratio (TM/TP), total methylxanthines (MX) and total flavanols (TF). In lines 291-293 authors wrote that “due to the structural similarities of procyanidins with epicatechin, a relative quantification of procyanidins was performed in terms of milligram equivalent of epicatechin per g of sample” – so in terms of above list of substances, how did you distinguish procyanidin B2 or other procyanidins on chromatogram to give values for PCA evaluation (Figure 1B)?  please provide information in the methodology

Conclusions - I have no comments; the conclusions are based on the results of the authors' research.

Author Response

Response to Reviewer 2 Comments

line 50-62 – this paragraph needs expansion in citations. Here I provide recommended articles: https://doi.org/10.1016/j.lwt.2018.08.030, https://doi.org/10.1016/j.foodres.2016.03.026, https://dx.doi.org/10.1021/acs.jafc.0c00412 , https://doi.org/10.1080/07373937.2016.1175470

Response 1: This version of the manuscript has been modified in this section, according to the instructions of the evaluator. The references most related to the document were added. See references 16 and 17.

Results and Discussion - authors need to discuss the findings with literature (chemical composition, influence of different harvest years, influence of different cultivar areas, influence of different clones). The current discussion of the results with literature is scarce.

Response 2: This version of the manuscript has been modified in this section, according to the recommendations of the evaluator. Particular conditions for the two regions involved in the study were included. Also, greater details in the chemometric analysis was provided, and some sentences were slightly modified for a better understanding of the readers

line 119-121 - please describe better with more details the principal component analysis (PCA). Are all analyzed parameters qualified for PCA analysis? What was the lowest correlation value of parameters with the generated first and second principal components?

Response 3: PCA was describe with more details, see item 2.1.

Figure 1 A and B – authors must enlarge numbers and codes of substances especially on figure 1B for better readability / or provide better resolution of figure like Figure 3.

Response 4: Numbers and codes of substances were enlarge.

Figure 2 – from which year of cocoa harvest are these values? 2019 or 2020 or both?, please give information.

Response 5: In the description of the figure 2, the information was provided.

Figure 3 – give information which harvest is from 2019 and which from 2020

Response 6: The years were specified in the figure 2.

For PCA and PLS-DA provide as supplementary materials figures with component 3 and 4 with commentary, because the Figure 1 explain only 55,3% of total variance and Figure 3 explain only 50,5% of total variance.

Response 7: Description for PCA with component 3 and 4 is at supplementary material. For PLS-DA it was not possible because the statistical program didn’t perform the analysis for components 3 and 4 (script out of bounds).

line 252-261 – authors must provide information how long gathered pods were stored and in what conditions before cocoa beans were separated from them and freeze-dried. Please provide also name of model and manufacturer of device used for grinding the beans. Whether the powdered samples were vacuum packed, modified atmosphere packed or otherwise. What packaging were these samples packaged in for storage? Did the authors mean -25C, not +25C?

Response 8: The information is provided in the item 3.2. Samples.

line 273 - Did the authors mean -20C, not +20C?

Response 9: It was corrected by -20°C.

line 276-294 – Please provide information how you obtained quantitative data about procyanidin B2 (PB2), procyanidins (D1 to D11), total monomers (TM), total procyanidins (TP), monomer/procyanidin ratio (TM/TP), total methylxanthines (MX) and total flavanols (TF). In lines 291-293 authors wrote that “due to the structural similarities of procyanidins with epicatechin, a relative quantification of procyanidins was performed in terms of milligram equivalent of epicatechin per g of sample” – so in terms of above list of substances, how did you distinguish procyanidin B2 or other procyanidins on chromatogram to give values for PCA evaluation (Figure 1B)?  please provide information in the methodology.

Response 10: We used a previously validated method (Ref 37), in that work we did the identification of some procyanidins an all the validation of the method in different cocoa samples. We included some information in the methodology section (Section: 3.4. Determination of the polyphenol and methylxanthine contents) and in the PCA results (Section: 2.1. Chemical composition of different cocoa clones).

Conclusions - I have no comments; the conclusions are based on the results of the authors' research.

Response 11: The authors welcome the reviewer's comments.

Round 2

Reviewer 2 Report

The authors addressed all comments and made the necessary corrections.